# LRP5, SLC6A3, and SOX10 Expression in Conventional Ameloblastoma

**DOI:** 10.3390/genes14081524

**Published:** 2023-07-26

**Authors:** Lorena Correa-Arzate, Javier Portilla-Robertson, Josué Orlando Ramírez-Jarquín, Luis Fernando Jacinto-Alemán, Claudia Patricia Mejía-Velázquez, Francisco Germán Villanueva-Sánchez, Mariana Rodríguez-Vázquez

**Affiliations:** 1Department of Oral Medicine and Pathology, Postgraduate Division, Dental School, National Autonomous University of Mexico, Mexico City 04510, Mexicojpr@unam.mx (J.P.-R.); dramejiavelazquez@outlook.es (C.P.M.-V.); 2Neurosciences Division, Cellular Physiology Institute, National Autonomous University of Mexico, Mexico City 04510, Mexico; 3Oral and Maxillofacial Pathology, Dental School, ENES-Leon, National Autonomous University of Mexico, Guanajuato 37684, Mexico; drvillanueva.enesunam@gmail.com; 4Infectomic and Molecular Pathogenesis Department, CINVESTAV, National Polytechnic Institute, Mexico City 07738, Mexico; marrova89@gmail.com

**Keywords:** gene expression profiling, ameloblastoma, bioinformatics, SOX10, LRP5, SLC6A3, microarray analysis

## Abstract

Cell proliferation and invasion are characteristic of many tumors, including ameloblastoma, and are important features to target in possible future therapeutic applications. Objective: The objective of this study was the identification of key genes and inhibitory drugs related to the cell proliferation and invasion of ameloblastoma using bioinformatic analysis. Methods: The H10KA_07_38 gene profile database was analyzed by Rstudio and ShinyGO Gene Ontology enrichment. String, Cytoscape-MCODE, and Kaplan–Meier plots were generated, which were subsequently validated by RT-qPCR relative expression and immunoexpression analyses. To propose specific inhibitory drugs, a bioinformatic search using Drug Gene Budger and DrugBank was performed. Results: A total of 204 significantly upregulated genes were identified. Gene ontology enrichment analysis identified four pathways related to cell proliferation and cell invasion. A total of 37 genes were involved in these pathways, and 11 genes showed an MCODE score of ≥0.4; however, only SLC6A3, SOX10, and LRP5 were negatively associated with overall survival (HR = 1.49 (*p* = 0.0072), HR = 1.55 (*p* = 0.0018), and HR = 1.38 (*p* = 0.025), respectively). The RT-qPCR results confirmed the significant differences in expression, with overexpression of >2 for SLC6A3 and SOX10. The immunoexpression analysis indicated positive LRP5 and SLC6A3 expression. The inhibitory drugs bioinformatically obtained for the above three genes were parthenolide and vorinostat. Conclusions: We identify LRP5, SLC6A3, and SOX10 as potentially important genes related to cell proliferation and invasion in the pathogenesis of ameloblastomas, along with both parthenolide and vorinostat as inhibitory drugs that could be further investigated for the development of novel therapeutic approaches against ameloblastoma.

## 1. Introduction

Although classified as benign, ameloblastoma is the odontogenic tumor with the most serious implications for the patient. Furthermore, it is probably the tumor with the greatest surgical controversy regarding its treatment, given its locally invasive behavior, high recurrence rate, and metastatic potential [1,2,3]. Conventional ameloblastomas tend to invade the underlying bone and soft tissue, presenting tumor islands far from the main tumor mass, which requires the removal of a wide margin of healthy tissue around the clinical limit to avoid recurrence, potentially leading to greater morbidity in the craniofacial complex as well as a loss of function and aesthetics [4]. Thus, conservative treatments such as enucleation or curettage are associated with a high recurrence rate. Understanding the etiology of this type of tumor could help to discover new therapeutic alternatives. At the molecular level, it has been suggested that the genetic processes involved in cytodifferentiation, invasion, and cell replacement are associated with the development of ameloblastomas; however, there are few studies in the existing literature that allow us to understand such processes in this tumor [5,6,7].

Gene expression profiling of tumor cell populations has improved our understanding of the pathogenesis of human tumors, as well as facilitating improved diagnosis at the molecular level [7]. The identification of specific genes or groups of genes that are deregulated—and, therefore, play an important role in the initiation, worsening, or determination of tumor morphology—could lead to the development of new diagnostic and therapeutic approaches. The microarray assay is a powerful tool for the identification of deregulated genes in tumor tissue, allowing for the subsequent identification of hundreds of differentially expressed genes involved in various molecular functions and biological processes through bioinformatic analysis, thus improving the efficiency and precision of the results obtained [8,9].

In this study, we identify the biological functions associated with differentially expressed genes in ameloblastomas and explore those related to cell proliferation and invasion through bioinformatic analyses in order to identify key genes. Furthermore, we discuss the potential of manipulating these genes for specific therapeutic purposes.

## 2. Materials and Methods

### 2.1. Microarray Data Information 

The H10KA_07_38_greater_2_Up microarray/gene profile database, located in the SciELO Data repository, is a free public database containing the gene expression profiles of conventional ameloblastomas [10]. Microarray data were acquired using a GeneChip Human Mapping 10K Array (CHIP H10KA_07_38, AFFYMETRIX, Santa Clara, CA, USA), taking 15 conventional ameloblastomas and 16 dental follicles (as the control). The H10KA_07_38_greater_2_Up considers a Z-score cutoff of >2.0, which was processed using the GenArise software for CHIP H10KA_07_38. This study was approved by the Institutional Technical Committee of the Support Program for Research and Technological Innovation Projects (DGAP/1956/2019/UNAM). 

### 2.2. Differentially Expressed Gene (DEG) Selection and Enrichment Analysis

To investigate global gene expression changes in ameloblastomas, the over-regulated and unregulated genes obtained from the H10KA_07_38 gene profile data sheet were submitted to the RStudio (Version 2021.09.2 + 382) software to determine the genes characterized by increases or decreases in Log FC. The H10KA_07_38_greater_2_Up genes were analyzed by gene ontology enrichment analysis (GO terms) using ShinyGO v0.76 Gene Ontology Enrichment Analysis in order to explore the following functional categories: molecular function (MF), biological process (BP), and cellular component (CC). Only statistically significant elements with *p* < 0.05 (FDR) that were related to cell proliferation and invasion were selected [11]. 

### 2.3. Protein–Protein Interaction (PPI) and Selection of Hub Genes

The retrieved significantly enriched genes were submitted to the Search Tool for the Retrieval of Interacting Genes (STRING version 11.0, available online: http://string-db.org/, accessed on 9 February 2023), in order to obtain a protein–protein interaction network [12]. The interaction network file was imported into the Cytoscape software to analyze the interactions with a combined score of >0.4. A plugin for molecular complex detection (MCODE) was used to detect the clustered modules in the PPI networks with MCODE score of degree cutoff = 2, node score cutoff = 0.2, k-core = 2, and max depth = 100. 

Only clustered genes with MCODE score > 0.4 were selected. To determine their clinical relevance and relationship with survival, Kaplan–Meier online analysis was conducted (KM plotter, http://kmplot.com/analysis, accessed on 9 February 2023) to estimate the overall survival by adjusting the follow-up threshold to 60 months concerning head and neck squamous carcinoma of overall survival, considering only genes with HR > 1 and *p* < 0.05 for consideration as hub genes [13]. 

### 2.4. Validation of Hub Genes by RT-qPCR

Five additional independent conventional ameloblastomas and three dental follicles (as control) were employed for total RNA extraction using an AllPrep DNA/RNA FFPE Kit (80234, Qiagen, Germany). Briefly, 50 μm slides of each sample were obtained, de-paraffinized, and total RNA was obtained following the manufacturer’s protocol and diluted in 30 μL of RNase-free water. To determine the RNA concentration and purity, a NanoDrop ND-2000 spectrophotometer (Thermo Fisher, Rochester, NY, USA) was employed, considering only samples with a 260/280 ratio of ≥1.8 [14]. RT-qPCR was performed using a QuantiNova SYBR Green RT-PCR Kit (Qiagen, Cat. 208152, Germany). All reactions were performed in triplicate on ABI PRIS 7000 Sequence Detection Systems (Waltham, MA, USA). Data were normalized with glyceraldehyde and the primer sequences were as follows: for LRP5, 5′-CCCAAACTGTCTGTCCTGGT-3′(sense) and 5′-CCCAGCTGTGCATCACTAGA-3′(antisense); for SOX10, 5′-GGCACAGTACCTGGCATTTT-3′(sense) and 5′-GGCCTCATGTTGTGTCCTTT-3′(antisense); for SLC6A3, 5′-GTGGCCTTTCAGACAGAAGC-3′ (sense) and 5′-AAGAGGGTGTGTCCTTGTGG-3′(antisense); and, for GAPDH, 5′-ACCACAGTCCATGCCATCAC-3′ (sense) and 5′-TCCACCACCCTGTTGCTGTA-3′ (antisense). Relative gene quantification was calculated using the 2^-(ΔΔCt) method [15]. For determination of significant gene expression, a paired *t*-test was performed, considering *p* < 0.05 with a 95% confidence interval to indicate a significant difference.

### 2.5. Immunohistochemistry Assay for Expression Validation

Five additional ameloblastomas and two dental follicles (as control) were obtained to determine the gene expression of SOX10, SLC6A3, and LRP5 by peroxidase immunohistochemistry assay. The slides were deparaffinized and rehydrated conventionally in xylene and alcohol washes. Antigenic retrieval was performed in 10 mM citrate buffer in a microwave histoSTATION at 100 °C for 5 min in GPR/20S histomodule (KOS Millestone, Sorisole, BG, Italy). Endogenous peroxidase nonspecific background blocking was performed, as reported previously [14]. The slides were incubated overnight at 4 °C with primary antibodies for SOX10 (sc-365692, Santa Cruz Biotechnology, Paso Robles, CA, USA), SLC6A3 (sc-32259, Santa Cruz Biotechnology, Paso Robles, CA, USA), and LRP5 (GTX64412, GeneTex, Irvine, CA, USA), all adjusted to concentrations of 1:200. Negative controls were produced by substitution of the primary antibody by PBS. The immunodetection results were visualized using an ImmunoDetector DAB HRP Brown Immunohistochemistry (Bio SB, BSB 0007, Goleta, CA, USA), following the manufacturer’s protocol. The slides were observed using a Leica DM750 microscope, obtaining five photomicrographs at 400× magnification from each sample by using a Leica ICC50 HD camera. The intensity of staining (optical density) was assessed using ImageJ software (NIH, Bethesda, Rockville, MD, USA), with the quantification calibrated to establish the scale of optical density at 0–0.9/negative, 1–1.9/mild, 2–2.9/moderate, and >3/intense. The cell positive proportion was obtained through a semiquantitative protocol, categorizing the proportion as (0) absent, (1) 1 to 10%, (2) 11 to 50%, and (3) greater than 50%. For SOX10, it is mandatory to consider nuclear immunoexpression as positive [16].

### 2.6. Bioinformatic Screening of Candidate Inhibitory Drug

Using the LINCS (Library of Integrated Network-Based Cellular Signatures) datasheet tools—specifically, Drug Gen Budger (https://maayanlab.cloud/DGB/ 23 February 2023)—the validated genes were targeted to identify drugs and small molecules that regulate gene expression. The downregulate list was obtained to select the principal drugs that presented significant *p* and q values (<0.05) and log FC < 1, and for which an inhibitory assay had been performed on a tumor or epithelial cell line [17]. A Venn diagram was generated to determine which drugs coincided with the inhibition of the target genes. Subsequently, the obtained drugs were analyzed in DrugBank Online (https://go.drugbank.com/, 23 February 2023) to estimate their application with respect to head, neck, and oral cancer treatments, searching all available clinical trials [18]. 

## 3. Results

### 3.1. Clinical Features

According to the clinical variables of our study groups, the mean age of the ameloblastoma group was 37.8 ± 17.7 years old, the distribution by sex was 9 males and 6 females, and the 15 specimens were all located in the mandible. The control group presented a mean age of 18.7 ± 10.7 years old, 75% were men, and 37.5% presented in the mandible.

### 3.2. DEG and Enrichment Analysis

We identified 252 differentially expressed genes, including 204 relatively upregulated genes and 48 relatively downregulated genes (Figure 1). Differentially expressed genes with the associated *p*-values and logFC are provided in Appendix A. Enrichment analysis of the differentially expressed genes revealed 62 enriched pathways, 4 of which were directly related to cell proliferation and cell invasion (locomotion (GO:0040011), cell migration (GO:0016477), cell motility (GO:0048870), and cell proliferation (GO:0033687); see Appendix A).

### 3.3. PPI Construction and Hub Genes Selection

A total of 37 genes were obtained from the related cell proliferation and cell invasion criteria. After Cytoscape-MCODE plug-in analysis, only 11 genes showed a score of ≥0.4 (Table 1; Figure 2A). The Kaplan–Meier plot survival analysis indicated that only SLC6A3, SOX10, and LRP5 presented negative associations with overall survival (HR = 1.49 (*p* = 0.0072), HR = 1.55 (*p* = 0.0018), and HR = 1.38 (*p* = 0.025), respectively; Figure 2B).

### 3.4. RT-qPCR Validation of Hub Genes in Independent Samples

The gene expression analysis of LRP5, SLC6A3, and SOX10 by RT-qPCR indicated relatively high-level expression (1.3, 3.1, and 2.2, respectively) in all independent samples obtained from patients (Figure 3). A paired *t*-test was conducted to compare the expression levels of these genes, which did not show any significant differences (*p* > 0.05).

### 3.5. Immunoexpression Analysis

The immunoexpression of LRP5 was positive in all analyzed samples, with moderate intensity in all ameloblastic tumoral cells. Interestingly, we observed immunoexpression in fibroblasts of tumoral stroma. SLC6A3 was positive with mild to moderate intensity in two of five samples in all ameloblastic cells. SOX10 showed cytoplasmic immunoexpression in all samples; therefore, it should be considered negative (Figure 4).

### 3.6. Bioinformatic Inhibitory Drug Selection

Only two drugs with specific inhibitory action with respect to LRP5, SLC6A3, and SOX10 were selected: parthenolide and vorinostat (Figure 5). The target cells for parthenolide are A375 and VCAP, with a mean dose and time of 10 µM and 24 h. As for vorinostat, the target cells are NPC, MCF7, and A375, the mean inhibitory time is 24 h, and the inhibitory dose range is 0.019–10 µM. The highest inhibitory fold change was −2.26 for LPR5 with vorinostat (Table 2).

The DrugBank analysis indicated that parthenolide has not been considered in any clinical trial related to head, neck, or oral cancer. Meanwhile, a total of five trials considering vorinostat have been reported (three for head and neck cancer and two for oral cancer; see Table 3).

## 4. Discussion

Conventional ameloblastoma, despite being a benign odontogenic tumor, has the potential to invade the surrounding tissues. Its asymptomatic growth and volume increase are related to bone destruction or cortical perforation, which may be detected through imaging. For these reasons, relevant treatments typically include wide excisions and facial reconstruction, considerably affecting the quality of life of the patient [3,4]. One of the characteristics that allows ameloblastomas to present these behaviors is the greater proliferative capacity of their cells; however, the mechanisms by which the tumor cells acquire this characteristic are still poorly understood [6,7]. Microarray analysis is a powerful and high-throughput tool for the global characterization of gene expression, and diverse reports have identified highly expressed genes with their hierarchical methods and bioinformatic analyses in ameloblastomas, suggesting that the identification of key deregulated genes and their potential roles in cellular proliferation and invasion could provide new therapeutic targets for the development of novel approaches [19,20,21,22]. The results of our functional enrichment analysis yielded multiple pathways involved in the regulation of transcription, proliferation, cell differentiation, and metabolic pathways, including glucose metabolism, and we decided to explore the pathways with the greatest significance in cell proliferation. Consequently, three genes were chosen from the enrichment analysis for further validation by RT-qPCR: SLC6A3, SOX10, and LRP5. These genes exhibited patterns similar to those in the microarray analysis. At the time of this study, there have been no previous reports on the functions of these genes in ameloblastomas; however, there have been reports of cell proliferation and invasion functions in other neoplasms.

SOX10 is a transcription factor that plays an essential role in the development and maturation of glia, which can activate the expression of myelin genes in oligodendrocytes, the nucleocytoplasmic expression of which is important for neural crest and peripheral nervous system development. High expression of this gene has recently been observed in salivary and ovarian tumors, as well as hepatocellular, nasopharyngeal, prostate, breast, and digestive carcinomas. Yin et al. suggested that SOX10 could promote the progression of bladder cancer by accelerating proliferation and invasion features, a significant inhibition or reduction of which could be obtained through the use of siRNA [23]. In another cell type similar to human hepatocellular carcinoma cells, Zhou et al. demonstrated that SOX10 expression was correlated with elevated levels of β-catenin [24]. β-catenin plays a central role in the Wnt signaling pathway, and its overexpression is known to result in neoplastic transformation, including ameloblastoma. Babichenko et al. and Santos et al. reported the relationship between the proliferative activity of ameloblastoma cells and the intranuclear localization of β-catenin, suggested to be correlated with Wnt/β-catenin signaling pathway, progression, and tumor recurrence [25,26]. The β-catenin protein is the main component of the Wnt/β-catenin pathway. When the normal cytoplasmic concentration is altered by inhibition of ubiquitin-proteasome-dependent degradation, the increase in cytoplasmic β-catenin concentration allows its entry into the nucleus and activates the transcription of genes related to cell proliferation. This process is strongly related to tumorigenesis as deregulation of the cadherin interaction may promote loss of cell adhesion, increasing the invasion potential. SOX10 overexpression may be an indirect marker of related Wnt/β-catenin cell proliferation and invasion potential; however, it is strongly possible that it is not the only molecule relevant to the tumorigenesis mechanism. LRP5 is a co-receptor in the Wnt/β-catenin pathway. Nie et al. showed that its overexpression in gastric cancer was positively associated with advanced clinical stages and poor prognosis [27]. These adverse features could be related to increased proliferation, invasiveness, drug resistance, and upregulation of aerobic glycolysis. LRP5 knockdown led to smaller volume and less-proliferative tumors, possibly through disruption of tumorigenic Wnt/β-catenin pathway signaling [28,29]. 

The SLC6A3 human dopamine transporter gene has been consistently implicated in several neuropsychiatric diseases; however, studies on the pathophysiology of tumorigenesis remain scarce. Some reports have shown that the expression level of SLC6A3 is significantly higher in gastric carcinoma, hepatocellular carcinoma, and renal cell carcinoma than in control tissues, and that it may be related to increased metastasis and proliferation [30,31,32,33]. Significant findings on its behavior in the presence of an inhibitor drug indicated that a dose-dependent decrease in cell proliferation could be obtained [33]. In ameloblastoma, the overexpression of SOX10, LRP5, and SLC6A3 may possibly be related to increased cell proliferation, local invasion, and high recurrence rates; as such therapeutic alternatives such as parthenolide and vorinostat may be employed as inhibitors of these genes. Our immunoexpression analysis demonstrated positivity for SLC6A3 and LRP5. These two molecules have not been previously related to ameloblastomas, but have been related to other neoplasms such as clear cell renal cell carcinoma and osteosarcoma [33,34]. SOX10 was negative in all our samples as we only observed cytoplasmic and non-nuclear immunoexpression, which is considered a necessary criterion for its categorization as positive [35]. However, this finding is consistent with a previous study, which reported that negative immunoexpression can be considered as a parameter of the immunohistochemical profile of ameloblastoma [36]. The overexpression observed in the RT-qPCR results suggests that SOX10 could be associated with the modulation or preservation of germ line cell potential at the genomic level [37]. Schrödter et al. reported that SLC6A3 could be considered as a potential biomarker of recurrence-free survival in clear cell renal cell carcinoma, showing that associated mRNA and immunoexpression in specific regions of renal tissues could be an important clinical feature, which may even be considered as a possible therapeutic target [33]. In our analysis, only three samples were positive with a certain level of intensity and mild proportion of positive cells, preserving the pattern of reduced mRNA to protein expression observed in immunohistochemistry as previously reported. LRP5, as a co-receptor for the Wnt pathway, has been shown to correlate significatively with metastasis and overall survival in osteosarcoma [34]. Our results suggest that the presence of this molecule reaffirms the importance of the Wnt pathway in the development of ameloblastomas and the possibility of exploiting this pathway as a therapeutic target. An interesting result was immunoexpression of LPR5 in tumoral stromal fibroblasts. This finding suggests that a parenchyma–stroma interaction is present. Chantravekin and Koontongkaew, through 3D-organotypic cultures and co-cultures, showed that ameloblastoma-associated fibroblasts could promote the proliferation and invasion of tumor cells by TGF-β activation [38]. Fuchigami et al., through a 3D co-culture model, proved that fibroblast-associated cells assist tumor cell invasion and promote a histologically similar follicular pattern [39]. These reports indicate that the growth and histological pattern of the tumor depend jointly on ameloblastic cells and fibroblasts and, therefore, the presence of LRP5 in both cell types should be considered when postulating a therapeutic strategy. The implications of LRP5 in the pathogenesis and treatment of ameloblastoma may be greater. It has been reported that activation of the Wnt/β-catenin pathway and IL-6/STAT3 signaling by LRP5 could promote neoplasm cells to acquire chemoresistance and the cancer stem cell phenotype, which should be considered when establishing chemotherapeutic approaches against ameloblastomas [40]. At present, the chemotherapeutic management of ameloblastoma is preferable in patients with metastases or multiple recurrences. To reduce these recurrences, identification of clinical–surgical factors, as well as the histological pattern and the complete bone resection through marginal or segmental osteotomy, could help in the development of better prognoses. When mutations in BRAFV600E and SMO are observed, the possibility of using specific inhibitors for chemotherapy opens. The joint use of dabrafenib/trametinib or only dabrafenib has led to a remarkable reduction in tumor volume [41,42,43]. 

Parthenolide is derived from the plant *Tanacetum parthenium*, employed as a herbal medicine for anti-inflammatory and antimigraine purposes. It has recently been reported that parthenolide may be useful in the treatment of cancer. A clinical trial reported in DrugBank was related to the relationship between parthenolide and allergic contact dermatitis (ClinicalTrials.gov Identifier: NCT00133341) [44]. Its anticancer activity is related to inhibition of NF-κB signaling, STAT3, modulation of the MDM2–HDAC1 complex, and reduction of ROS [45]. There have been no reports related to its employment in ameloblastomas; however, in oral squamous cell carcinoma, it was found to promote apoptosis [46,47]. Vorinostat is an inhibitor of class I and II histone deacetylases. In 2006, it was the first histone deacetylase inhibitor employed against T-cell lymphoma and other malignancies [47]. Many anticancer mechanisms related to vorinostat have been reported, such as increasing the number of apoptotic cells, induction of autophagy, cell cycle arrest, inhibition of proliferation and migration, and increasing E-cadherin expression [48,49]. The DrugBank analysis confirmed the use of vorinostat for the treatment of neoplasms including oral, head, and neck cancer; however, no evidence specifically for ameloblastoma has been reported yet. Molecular target treatments are still limited, and the evidence reported has focused on BRAF and/or MEK inhibitors [42,43]. Both parthenolide and vorinostat have been shown to be effective in combined use with other chemotherapeutic drugs, as they can help reduce the associated toxicity, as well as chemoresistance, thus amplifying the therapeutic effect of chemotherapy, in cholangiocarcinoma and T-cell acute lymphoblastic leukemia cell lines [50,51].

Advances in the field of bioinformatics provide us with valuable information on the molecular functions of genes, in order to understand the pathogenesis of tumors and, thereby, develop novel therapies. From our bioinformatic analysis, we observed that vorinostat presented an important inhibitory fold change with respect to LPR5; therefore, when considering the potential of this molecule regarding its participation in the Wnt/β-catenin pathway, this drug could be postulated as a good alternative or adjuvant. The information obtained from bioinformatics platforms provides us with a more accurate approach to the preclinical phase of clinical trials, which could reduce costs and experiments. However, despite being a powerful guide, the suggestion is that these new therapies through both drugs must be verified in controlled in vitro and in vivo environments, with close follow-up to guarantee their efficacy in affected patients.

## 5. Conclusions

We identified LRP5, SLC6A3, and SOX10 as potentially important genes related to the cell proliferation and invasion of ameloblastomas through a large-scale gene expression analysis using a microarray approach, validating the expression results through immunohistochemical assays. Consequently, we proposed the use of two drugs (parthenolide and vorinostat) for possible inhibitory treatment. Although more research will be necessary to clarify the molecular pathways of these genes in relation to ameloblastoma tumorigenesis, our study provides a basis and a possible pharmacological treatment for further research, which is expected to contribute to improving the prognostic factors for ameloblastoma.

## Figures and Tables

**Figure 1 genes-14-01524-f001:**
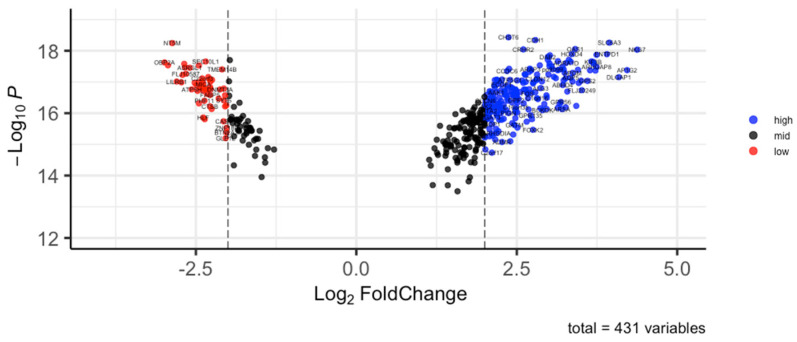
Ameloblastoma DEGs. The volcano plot obtained from RStudio analysis data shows 204 relatively upregulated genes and 48 relatively downregulated genes.

**Figure 2 genes-14-01524-f002:**
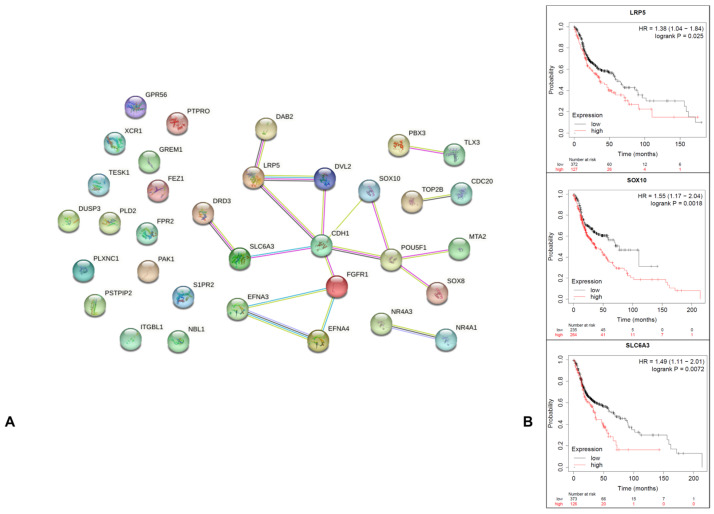
PPI and Kaplan–Meier survival plots: (**A**) A total of 37 differentially expressed genes were identified in the network; (**B**) prognostic information for the 3 core genes. The Kaplan–Meier plotter online tool was used to analyze the prognostic information, and only LRP5, SLC6A3, and SOX10 were found to be significantly (*p* < 0.05) associated with survival rate (overall survival of 60 months) with HR > 1, suggesting that their overexpression is related to poor prognosis [13].

**Figure 3 genes-14-01524-f003:**
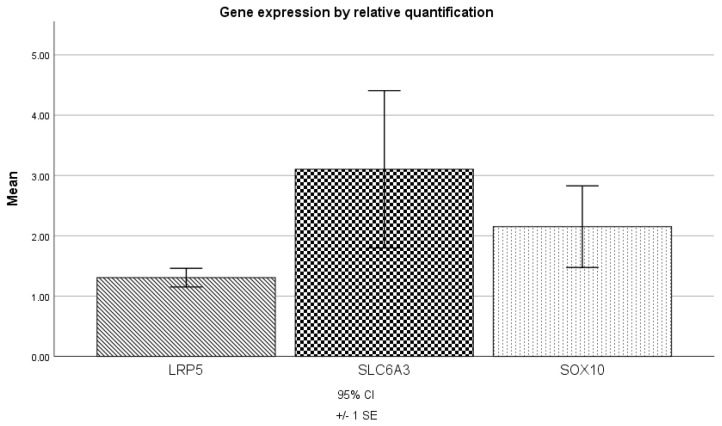
Gene expression analysis by RT-qPCR. SLC6A3 and SOX10 presented overexpression level > 2. The *t*-test did not reveal any significant differences between the analyzed genes.

**Figure 4 genes-14-01524-f004:**
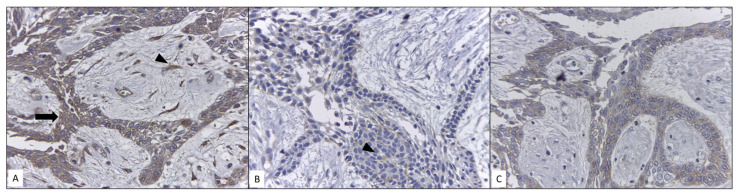
Immunoexpression of LRP5, SLC6A3, and SOX10: (**A**) LRP5 shows intense cytoplasmic immunoexpression in all ameloblastic tumor cells (arrow) and fibroblastic cells of stroma (arrowhead); (**B**) SLC6A3 shows mild cytoplasmic immunoexpression in ameloblastic cells (arrowhead); and (**C**) SOX10 shows cytoplasmic immunoexpression. Photomicrographs at 400×.

**Figure 5 genes-14-01524-f005:**
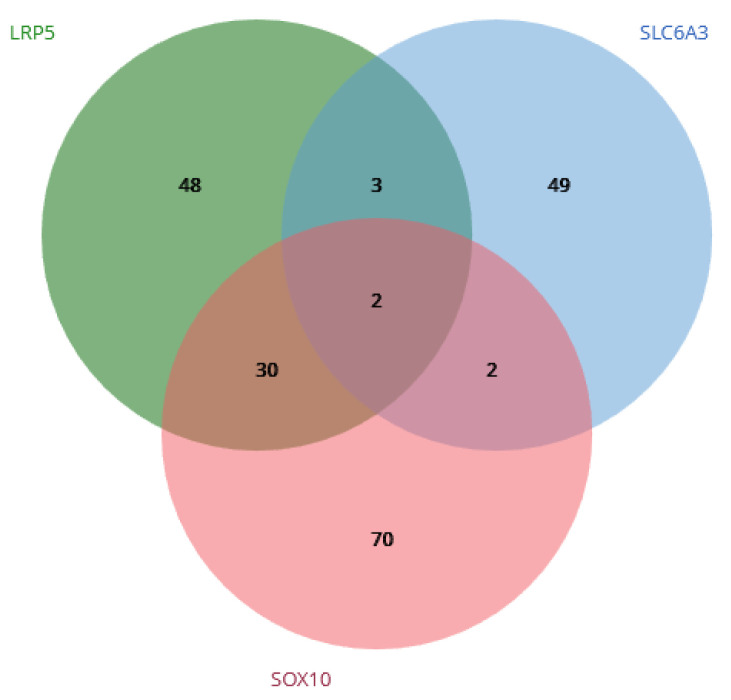
Venn diagram from Drug Gen Budger showing the downregulate list for the LRP5, SOX10, and SLC6A3 genes. Only two drugs presented an inhibitory effect for these three genes: parthenolide and vorinostat.

**Table 1 genes-14-01524-t001:** Cytoscape-MCODE plug-in analysis.

Genes with MCODE index > 0.4	DVL2, LRP5, CDH1, POU5F1, PBX3, CDC20, SLC6A3, EFNA3, FGFR1, SOX10, EFNA4
Genes with MCODE index < 0.4	SOX8, TOP2B, TESK1, DUSP3, NR4A3, NR4A1, PLD2, PLXNC1, PAK1, MTA2, FEZ1, PTPRO, DRD3, PSTPIP2, DAB2, NBL1, TLX3, FPR2, XCR1, ITGBL1, ADGRG1, S1PR2, GREM1, DAB2, FPR2, PLXNC1

**Table 2 genes-14-01524-t002:** Bioinformatic inhibitory drug selection for SOX10, LPR5, and SLC6A3.

Target Gene	Drug Name	LINCS sig_id	Cell Line	Time	Dose	*p*-Value	q-Value	Fold Change	Specificity
lrp5	parthenolide	CPC006_A375_24H:BRD-K98548675-001-02-6:10	A375	24 h	10.0 µM	1.1156 × 10^−11^	1.5111 × 10^−10^	−1.42111	0.00017446
slc6a3	parthenolide	CPC013_VCAP_24H:BRD-K28120222-001-05-0:10	VCAP	24 h	10.0 µM	0.00019181	0.00097252	−1.0578	0.00016545
sox10	parthenolide	CPC006_A375_24H:BRD-K98548675-001-02-6:10	A375	24 h	10.0 µM	1.3492 × 10^−9^	1.1894 × 10^−8^	−1.69409	0.00017446
lrp5	vorinostat	CPC016_NPC_24H:BRD-K81418486:10	NPC	24 h	10.0 µM	2.4155 × 10^−14^	4.2882 × 10^−12^	−2.26067	0.00019996
slc6a3	vorinostat	HDAC002_MCF7_24H:BRD-K81418486-001-10-3:0.0195312	MCF7	24 h	0.0195312 µM	4.3444 × 10^−6^	5.8744 × 10^−5^	−1.57292	0.00016504
sox10	vorinostat	CPC006_A375_24H:BRD-K81418486:10	A375	24 h	10.0 µM	3.4397 × 10^−14^	4.6197 × 10^−13^	−2.12695	0.00014943

**Table 3 genes-14-01524-t003:** Clinical trials reported in DrugBank for vorinostat.

Search Criteria	Phase	Status	Purpose	Clinical Trial.Gov Identifier
Oral cancer	2	Completed	Treatment	NCT01175980
Oral cancer	1	Terminated	Treatment	NCT01249443
Head and Neck cancer	2	Active, not recruiting	Treatment	NCT04357873
Head and Neck cancer	1, 2	Active, not recruiting	Treatment	Not available
Head and Neck cancer	Not available	Completed	Basic Science	NCT00735826

## Data Availability

The supporting data for reported results can be found in SciELO Data H10KA_07_38_greater_2_Up, available at: https://data.scielo.org/file.xhtml?persistentId=doi:10.48331/scielodata.Z2S8X9/ZWJGGP&version=1.0 (accessed on 9 February 2023).

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
