# Peer review of "LRP5, SLC6A3, and SOX10 Expression in Conventional Ameloblastoma"

_genes, 2023, doi:10.3390/genes14081524_

Round 1

Reviewer 1 Report

Correa-Arzate and colleagues used a gene expression profile database of 15 ameloblastomas and employed various bioinformatics tools to find Differential Gene Expression compared to dental follicle. Three genes showed a negative correlation with overall survival (SLC6A3, SOX10, and LRP5) and its expression were validated by RT-qPCR. Based on bioinformatics analysis, parthenolide and vorinostat were identified as potential inhibitory drugs targeting the aforementioned three genes.

In conjunction with genetic mutation studies, gene expression analysis plays a crucial role in unraveling the underlying mechanisms of neoplastic lesions. This approach has been extensively employed across various types of lesions, including ameloblastoma, to gain valuable insights into their pathogenesis. In a serious effort to increase the impact of the discoveries, Correa-Arzate and colleagues aimed to validate the expression levels of SLC6A3, SOX10, and LRP5 in additional samples. However, only a small number, specifically five lesions, were used. A statistical test with the RT-qPCR results were not performed. It would be attracting to explore the potential application of the suggested inhibitory drugs in cultured cells. Performing in vitro experiments using ameloblastoma cultured cells (or even cell lines from other cell types such as HaCaT) could provide valuable insights into the efficacy and mechanisms of action of these drugs on cell proliferation.

Furthermore, it is recommended that the authors conduct a comprehensive search for other studies exploring the gene expression profile or immunohistochemistry related to the genes SLC6A3, SOX10, and LRP5 in ameloblastoma. This would help to validate these targets and their potential implications.

The description of materials and methods (Validation of hub genes by RT-qPCR section) is comprehensive; however, it is important to avoid verbatim replication of the manufacturer's manual. Instead, aim to provide a concise account of the experimental procedures. Reference 14 in this section is a self-reference that is not appropriately used, as the article does not provide additional information about the RT-qPCRmethod.

It is necessary to review the English text as certain sections do not appear to have been translated effectively from Spanish. For instance: page 5 line 195:
… “SLC6A3, SOX10 y LRP5 by RT-qPCR”

… “(2.5, 2.1 y 1.3 respectively)”

Author Response

Thank you very much for your comments, they were very proactive and helped us greatly to improve the manuscript. We list the modifications made:

  1. However, only a small number, specifically five lesions, were used.
    1. The incidence of ameloblastoma in histopathological diagnostic centers from universities like ours, is not very high related to COVID-19 pandemic consequences, they have gradually begun to recover their concentration potential for this and other neoplasms, as far as possible we increase the number of new cases for the immunohistochemical assay to establish an independent validation process.
  2. A statistical test with the RT-qPCR results were not performed.
    1. Thanks for the observation. a paired t-test was performed.
  3. It would be attracting to explore the potential application of the suggested inhibitory drugs in cultured cells. Performing in vitro experiments using ameloblastoma cultured cells (or even cell lines from other cell types such as HaCaT) could provide valuable insights into the efficacy and mechanisms of action of these drugs on cell proliferation.
    1. Thank you very much for the observation and suggestion to use a cell line such as HaCaT (we tried to get it in Mexico, but we couldn't). This manuscript represents a part of a larger project, in which obtaining primary cell lines of ameloblastoma was planned, however, to date in our primary cultures, we have observed low or no adherence and a large amount of cell lysis in these cell line, for which we are still trying to obtain a stable cell line to take the next step, for which we regret not being able to fulfill this particular request, knowing this objective limitation previously the Drug Bank bioinformatics tool was used to determine if the drug of interest had any therapeutic use in a clinical trial (page 7, line 246-248).
  4. Furthermore, it is recommended that the authors conduct a comprehensive search for other studies exploring the gene expression profile or immunohistochemistry related to the genes SLC6A3, SOX10, and LRP5 in ameloblastoma. This would help to validate these targets and their potential implications.
    1. Thanks for the suggestion, the IHC assay was incorporated.
  5. The description of materials and methods (Validation of hub genes by RT-qPCR section) is comprehensive; however, it is important to avoid verbatim replication of the manufacturer's manual. Instead, aim to provide a concise account of the experimental procedures.
    1. The section was revised reducing the replication of the manufacturer's protocol.
  6. Reference 14 in this section is a self-reference that is not appropriately used, as the article does not provide additional information about the RT-qPCR method.
    1. We apologize for the fact, we add one more reference that provides more detail on the 2delta ct quantification analysis used.
  7. Translation errors were corrected.

Reviewer 2 Report

The authors identified LRP5, SOX10, SLC6A3 that could be related to cell proliferation and invasion of ameloblastoma. To identify the potential genes, the authors first profiled the gene expressions in both dental follicles (control) and conventional ameloblastomas. Second, differential gene expression analysis and enrichment analysis were conducted to select potential candidates that related to cell proliferation and invasion. Combined with survival analysis, LRP5, SOX10, and SLC6A3 are ranked as top candidates. The manuscript was well written, and the findings are clear.

Minor points:

1.      Figure 2: What’s the data source used in this study for the survival analysis? Please explicitly reference the data source.

2.      Section 3.4, line 195: the authors stated that the RT-qPCR validation has been conducted in another group of independent samples, but there’s no figure showing these results. Please put the RT-qPCR results in the manuscript.

3.      Line 196, 197: the ‘y’ should be replaced by ‘and’

Author Response

Thank you very much for your comments, they were very proactive and helped us greatly to improve the manuscript. We list the modifications made:

  1. Figure 2: What’s the data source used in this study for the survival analysis? Please explicitly reference the data source.
    1. Thanks for the suggestion, the data was added in figure text.
  2. Section 3.4, line 195: the authors stated that the RT-qPCR validation has been conducted in another group of independent samples, but there’s no figure showing these results. Please put the RT-qPCR results in the manuscript.
    1. Thanks for the observation, the correction was made and added the pertinent statistical analysis and an illustrative figure.
  3. Line 196, 197: the ‘y’ should be replaced by ‘and’
    1. Sorry for the omission

Round 2

Reviewer 1 Report

Materials and Methods section still appears to be excessively lengthy and should be rewritten for conciseness. In the RT-qPCR, when describing RNA extraction, it is sufficient for authors to refer to the manufacturer's protocol instead of providing detailed explanations such as incubation time and buffer volumes used, as this method is commonly used and well-established. The same for the immunohistochemistry method. Authors should mention only modifications or special considerations regarding the materials, referring to established protocols or manufacturer's instructions for commonly used techniques, rather than providing an exhaustive description of every step. 

Author Response

Thanks for the observation, the methodology corresponding to the RT-qPCR and IHC assays were reduced, keeping only necessary features to explain the results. We are kind to your comments.